# HIGHER-ORDER GRAMMAR REPRESENTATIONS FOR MOLECULAR GENERATION AND LEARNING

**Yiming Huang**[1]    **Yujie Zeng**[2]    **Vijay Prakash Dwivedi**[3]    **Simone Foti**[1]
**Jianmin Wang**[4]    **Jure Leskovec**[3]    **Tolga Birdal**[1]
[1]Imperial College London,    [2]Queen Mary University of London,
[3]Stanford University,    [4]Yonsei University

## ABSTRACT

Molecular learning models are strongly shaped by their underlying representations, yet standard sequential and graph representations struggle to explicitly encode higher-order molecular topology such as ring systems and motifs. To address this gap, we introduce a Higher-order Grammar Representation (HGR), a principled, topology-aware framework that lifts molecules to combinatorial complexes and parses them into a compact sequence of production rules under a context-free higher-order grammar. We devise three complementary lifting strategies, which induce distinct grammars that balance topological expressiveness with rule compactness. To mitigate evaluation biases in existing benchmarks, we construct RingDiv, a ring-enriched benchmark of 1.18M molecules, and curate a higher-quality 300k subset. Across de novo molecular generation, HGR-based models achieve 100% validity and outperform strong SMILES- and diffusion-based baselines. We further develop a molecular foundation model that integrates HGR with higher-order structural inductive biases, yielding robust and transferable representations across downstream benchmarks.

## 1 INTRODUCTION

The effectiveness of molecular machine learning fundamentally depends on how we represent chemical structures (Huang & Von Lilienfeld, 2016). A desired representation should be expressive enough to capture the structural determinants of molecular behavior (Wu et al., 2024), *yet* constrained enough to support reliable generation, efficient learning, and transfer across tasks. Despite substantial progress, widely used representations struggle with one recurring bottleneck: molecules exhibit rich higher-order topology that is not naturally or explicitly encoded by standard atom and bond formalisms (Rottach et al., 2025). Higher-order structural primitives, including ring systems, conjugated subgraphs and functional motifs, are ubiquitous in chemistry (Ertl et al., 2025), and serve as primary determinants of a molecule's characteristics. Such dependencies are not reducible to independent pairwise edges (Battiston et al., 2020). This gap becomes particularly acute for generative modeling and for foundation models that aim to learn shared, transferable, compositional representations from large corpora.

Current molecular representations fall into two broad categories, both of which prove inadequate for capturing higher-order topology. *(i)* String representations, such as SMILES (Weininger, 1988), are compact and human-readable but syntactically fragile for generation, linearizing molecules in ways that only implicitly encode topology. *(ii)* Graph-based models treat molecules as networks whose inductive bias is fundamentally pairwise, limiting expressiveness and suffering from training pathologies that weaken long-range and global structural reasoning (Alon & Yahav, 2021). More recent higher-order encodings improve expressiveness (Huang et al., 2024; Papamarkou et al., 2024) but introduce structural conflicts across topological dimensions and significantly enlarge the combinatorial search space for generation. This representational gap has profound consequences: it limits our ability to build higher-order generative models that reliably produce chemically valid structures, hampers the development of foundation models that can learn and transfer compositional chemical knowledge, and ultimately constrains our capacity to computationally explore and optimize molecular space for applications in drug discovery, materials science, and chemical synthesis.

**Existing gaps.** The core challenge lies in satisfying competing requirements: *(i)* representations must simultaneously encode higher-order topological relations, *(ii)* preserve hierarchical compositionality, *(iii)* remain flexible for generation and learning with standard architectures, and *(iv)* allow exact recovery of the original molecule. Existing methods sacrifice one or more of these requirements. For instance, string methods give implicit topology to enable sequential processing (Wu et al., 2024; Mastrolorito et al., 2025). Similarly, pairwise graphs trade off expressiveness for computational efficiency, while higher-order topological methods sacrifice generation tractability for expressive power. Fragment-based vocabularies offer a promising middle ground, improving interpretability and sample efficiency by treating functional groups as semantic units (Erlanson et al., 2016; Guo et al., 2022). However, treating motifs as flat tokens captures only one slice of higher-order structure (Jin et al., 2018). Real molecules are hierarchical: atoms form substructures, substructures compose scaffolds and ring systems, and these components interact to determine global topology (Wan et al., 2025). Without an explicit mechanism to model this hierarchy, models can overfit local validity while assembling globally implausible structures, or preserve coarse scaffolds at the expense of chemically faithful local detail. Furthermore, a practical challenge in any grammar-based approach is rule explosion driven by complex boundaries and ring-rich chemistry (Kajino, 2019; Sun et al., 2025), which can destabilize both learning and generation. An additional data-level bottleneck compounds these modeling challenges: commonly used benchmarks underrepresent complex ring systems, which can mask meaningful differences between generative models and provide overly optimistic assessments of model capabilities in topologically diverse chemical space.

In this work, we propose **Higher-order Grammar Representations (HGR)**, a principled, lossless, and explicitly hierarchical representation that turns higher-order molecular topology into 'tokens' that can be used in a sequence modeling framework, while addressing the aforementioned research gaps. Our contributions are summarized as:

1. The central idea of HGR is to *(i)* lift a molecular graph into a combinatorial complex whose cells correspond to hierarchical and higher-order (multi-atom) relations, then *(ii)* parse each molecular complex into a sequence of production rules via a bottom-up contraction procedure, thereby inducing a higher-order grammar that supports exact decoding.

2. To address rule explosion, we introduce three lifting variants trading off motif granularity and boundary simplicity: frequency-guided merging, anchor-resolving boundary reduction, and ring-system guided hierarchical stabilization.

3. To better evaluate models on topologically complex molecules, we construct RingDiv, a ring-enriched benchmark that mitigates the dataset bias present in commonly used benchmarks and provides broader coverage of diverse ring systems.

We demonstrate HGR's utility through a grammar-constrained VAE (GVAE) for de novo generation and a HGR-based foundation model that learns from rule sequences alone. Our evaluation yields three key insights: *(i)* the higher-order grammar-based framework guarantees validity and supports efficient generation; *(ii)* the proposed representation proves universal, enabling a single pretrained model to support multiple tasks and improving transfer on downstream property prediction; and *(iii)* experimental results suggest that explicitly encoding hierarchical higher-order topology provides key inductive biases that benefit both generation and representation learning.

## 2 HIGHER-ORDER GRAMMAR REPRESENTATION (HGR)

**Higher-order complex and lifting operation.** A canonical representation of a molecule is a graph $\mathcal{G} = (\mathcal{V}, \mathcal{E})$ consisting of a vertex set $\mathcal{V}$ and an edge set $\mathcal{E}$. However, molecules often contain complex multi-way relations such as rings, functional groups, and other higher-order motifs that cannot be fully captured by pairwise edges (Battiston et al., 2020). To capture such higher-order structure, we lift molecular graphs into combinatorial complexes (CCs) (Hajij et al., 2022), a general formalism that unifies graphs, simplicial and cell complexes, and hypergraphs within a single mathematical framework. Mathematically, a CC is defined as a triple $\mathcal{C} = (\mathcal{V}, \mathcal{X}, \mathrm{rk})$, where $\mathcal{V}$ is a finite set of entities (also referred to as vertices), $\mathcal{X} \subseteq \mathrm{Pow}(\mathcal{V}) \setminus \{\emptyset\}$ is a collection of non-empty subsets of $\mathcal{V}$, called cells or relations, and $\mathrm{rk} : \mathcal{X} \to \mathbb{Z}_{\geq 0}$ is a rank function. This structure satisfies two conditions: for every vertex $v \in \mathcal{V}$, the singleton $\{v\}$ is included in $\mathcal{X}$ with $\mathrm{rk}(v) = 0$; and the rank function is order-preserving, meaning that for any $x, y \in \mathcal{X}$, if $x \subseteq y$, then $\mathrm{rk}(x) \leq \mathrm{rk}(y)$.

A cornerstone of Topological Deep Learning (TDL) is lifting, which transforms graph-structured data to more expressive topologies. We adopt a data-driven lifting strategy based on a motif vocab-

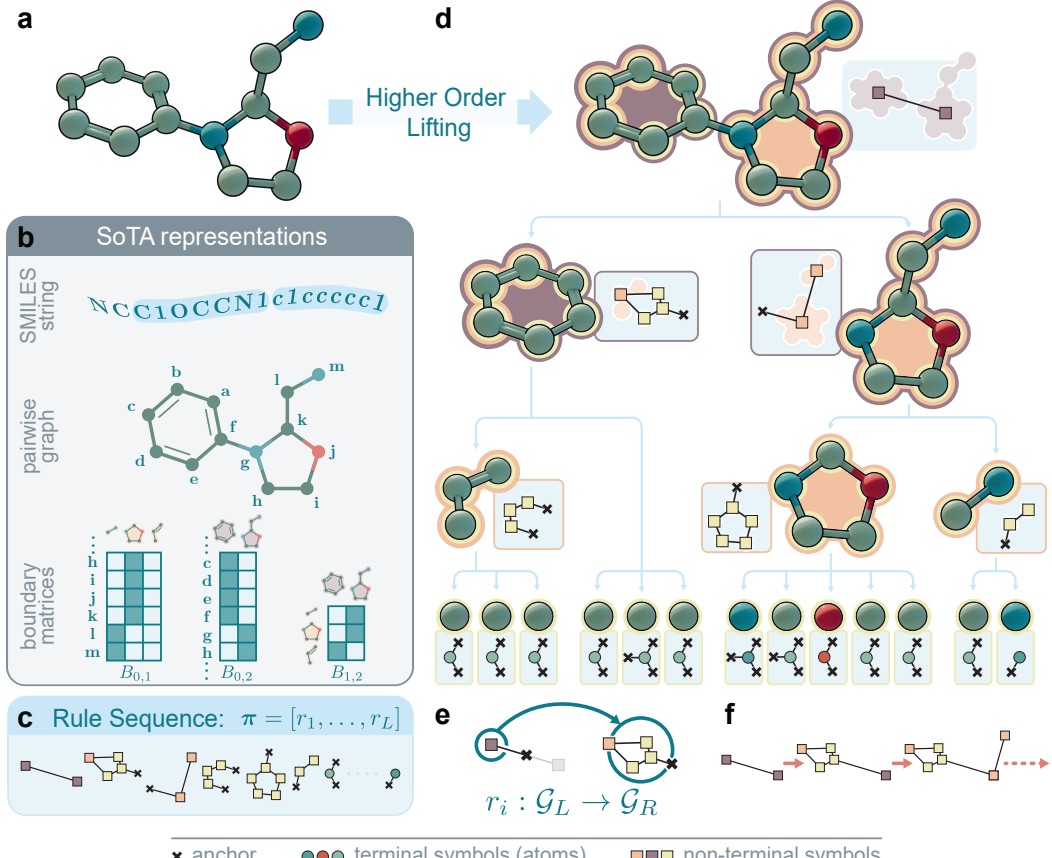

Figure 1: **Overview of molecular representations and our HGR framework.** **a**, Example molecule, which can be lifted to the combinatorial complex shown at the top of **d**. **b**, Classical state-of-the-art (SoTA) representations. **c**, Higher-order Grammar Representation (HGR): a molecule is serialized into a rule sequence $\pi = [r_1, \cdots, r_L]$, where each token corresponds to a fundamental structural unit. **d**, The full rule sequence compactly encodes the molecule's hierarchical higher-order topology. Each cell is associated with a rule token that indexes a shared rule corpus, enabling decomposition into reusable structural units. **e**, Rule specification. **f**, Derivation process: rules iteratively expand anchored non-terminals, progressively assembling the complex in **d**.

ulary mined from a molecule library. The resulting complex is then used to induce a higher-order molecular grammar. To ensure invertibility, we record the attachment interface of each production as an ordered list of anchors (external neighbours of the motif). This ordering disambiguates attachments during decoding and enables exact reconstruction of the original molecular complex $\mathcal{C}$. However, in merging-based lifting, boundary complexity becomes a major driver of rule growth: motifs with many anchors admit many distinct attachment interfaces and thus rapidly inflate the rule corpus. To balance expressivity and rule compactness, we instantiate three lifting variants trading off motif granularity and boundary simplicity: (1) MEG lifting serves as the frequency-guided merging baseline. (2) Building on MEG lifting, MIG lifting limits anchor proliferation by resolving complex boundaries via best-first motif expansion. (3) RSG lifting further stabilises lifting in cyclic chemistry through ring-system–aware merging, mitigating rule inflation from complex ring attachments.

Notably, each resulting CC admits an intuitive representation as a hierarchical tree (Fig. 1**d**), wherein each node corresponds to a production rule. Rather than representing molecules as flat strings, HGR encodes each molecule as a multi-scale rule sequence spanning atomic, mesoscopic motif, and whole-molecule levels. Leaf nodes represent atomic micro-structures, while internal nodes correspond to higher-order meso-structures. We then sort nodes by atom indices and perform a depth-first traversal to linearize the tree into a rule sequence.

**Higher-order grammar induction and derivation.** Concretely, HGR linearizes the inherent hierarchy of a molecular complex by recording a bottom-up contraction history, yielding a rule sequence that can be inverted to uniquely reconstruct the original complex. Formally, the grammar is defined as $\Gamma = (\Sigma, \mathcal{N}, S, \mathcal{P})$, where $\Sigma$ and $\mathcal{N}$ denote the finite sets of terminal and non-terminal symbols, respectively, $S \in \mathcal{N}$ is the start symbol, and $\mathcal{P} = \{r_1, r_2, \ldots, r_{|\mathcal{P}|}\}$ represents the corpus of production rules. We induce $\Gamma$ via a bottom-up parsing procedure based on progressive coarsening. At

each step, we contract a lowest-rank cell into a fresh non-terminal symbol and create a production rule that records both the contracted region and its anchor interface. The rule $r_i$ is then added to the corpus $\mathcal{P}$, and the complex is updated with fewer cells. After convergence, each molecular complex $\mathcal{C}_i$ is parsed into a rule sequence $\boldsymbol{\pi}^{(i)} = [r_1^{(i)}, r_2^{(i)}, \cdots, r_{L_i}^{(i)}]$, which records the sequence of contractions applied during parsing. Grammar induction proceeds incrementally alongside parsing, dynamically evolving as new patterns emerge and stabilizing once all molecular complexes are fully processed. The mapping from resulting rule sequences to molecular complexes is deterministic and lossless: the reverse application of $\boldsymbol{\pi}^{(i)}$ starting from $S$ uniquely reconstructs the original molecular complex, defining the derivation process $S \Rightarrow^{\boldsymbol{\pi}^{(i)}} \mathcal{C}_i$.

Without loss of generality, we detail the parsing and rule construction process for a single combinatorial complex $\mathcal{C} = (\mathcal{V}, \mathcal{X}, \mathrm{rk})$. Alongside the cell family $\mathcal{X}$, we maintain the underlying molecular graph $\mathcal{G} = (\mathcal{V}, \mathcal{E})$, i.e., the 1-skeleton that records atom types and covalent bonds. At each iteration, we select a cell $\sigma \in \mathcal{X}$ with the lowest rank and contract it into a fresh non-terminal symbol $N^*$, updating both $\mathcal{C}$ and $\mathcal{G}$ accordingly. When multiple adjacent cells share the same rank, we break ties at random and record their shared vertices. Since singleton cells have rank 0, parsing proceeds from atomic cells upward.

Each rule $r_i$ is defined by a compact interface graph on the left-hand side and a local neighbourhood graph on the right-hand side, i.e., $r_i : \mathcal{G}_L \to \mathcal{G}_R$. Specifically, $\mathcal{G}_L$ is a star-shaped interface graph

$$\mathcal{G}_L = (\{N^*\} \cup \mathcal{V}_{\mathrm{anc}}^\sigma, \{(N^*, v) \mid v \in \mathcal{V}_{\mathrm{anc}}^\sigma\}), \tag{1}$$

where incident edges preserve the ordered anchor interface. Anchors are the external neighbours of $\sigma$ in the skeleton graph, i.e., $\mathcal{V}_{\mathrm{anc}}^\sigma = \{v \in \mathcal{V} \setminus \sigma \mid \exists u \in \sigma \text{ s.t. } (u, v) \in \mathcal{E}\}$. The right-hand side $\mathcal{G}_R$ is the induced local subgraph over the contracted region and its anchors,

$$\mathcal{G}_R = (\sigma \cup \mathcal{V}_{\mathrm{anc}}^\sigma, \mathcal{E}^{\sigma \cup \mathcal{V}_{\mathrm{anc}}^\sigma}), \tag{2}$$

where $\mathcal{E}^{\sigma \cup \mathcal{V}_{\mathrm{anc}}^\sigma} = \{(u, v) \in \mathcal{E} \mid u, v \in \sigma \cup \mathcal{V}_{\mathrm{anc}}^\sigma\}$. Consequently, $\mathcal{G}_R$ captures both internal bonds within $\sigma$ and boundary bonds connecting $\sigma$ to its anchors. After creating $r_i$, we replace $\sigma$ with $N^*$ in the complex so that it can participate in subsequent contractions, and we continue until the structure reduces to a single non-terminal symbol, which we designate as the start symbol $S$.

Given the induced grammar $\Gamma$, a molecular complex can be reconstructed by sequentially applying the production rules in reverse. Starting from the start symbol $S$, we iteratively match the left-hand-side interface $\mathcal{G}_L$ in the current complex and replace it with the rule's right-hand side $\mathcal{G}_R$, until no non-terminals remain. To determine whether a production rule $r_i$ is applicable at each step, we use subgraph matching (Gentner, 1983) to test whether the current graph contains a subgraph that is isomorphic to $\mathcal{G}_L$. Since these patterns are typically small in scale, the matching process is efficient in practice. The resulting derivation $S \Rightarrow^{\boldsymbol{\pi}^{(i)}} \mathcal{C}_i$ provides a lossless reconstruction of the molecular complex, and structural dependencies can be explicitly preserved regardless of serialization order by employing positional encodings tailored for tree structures.

Notably, while the shift from sequential representations like SMILES to graphs has dramatically advanced molecular ML, HGR takes a further leap by explicitly incorporating hierarchical higher-order structures. The corresponding molecular graphs can be readily obtained by extracting the underlying skeletons from these complexes. Moreover, by generating these complexes directly, the framework naturally captures molecular "generation trajectories", thereby transcending conventional generation paradigms. Beyond these merits, HGR offers several additional advantages: each production rule corresponds to a specific higher-order motif, providing interpretability and structural insight; grammar induction from real molecular complexes inherently preserves valency and bonding constraints, ensuring chemical validity; and the representation supports exact reconstruction while generalising beyond the training distribution, enabling both interpolation and extrapolation.

## 3 EXPERIMENTS

### 3.1 DE NOVO MOLECULAR GENERATION WITH HGR

As a core task in drug discovery, de novo molecular generation provides novel candidates and directly determines the quality of resulting compounds. However, current molecular generative models predominantly rely on sequence- or graph-based representations that emphasize local chemistry

Table 1: De novo generation performance comparison. Top results are highlighted in **bold**.

| Method | QM9 (Ramakrishnan et al., 2014) | | | | | ZINC250k (Irwin et al., 2012) | | | | |
|---|---|---|---|---|---|---|---|---|---|---|
| | Val.↑ | Uni.↑ | Nov.↑ | FCD↓ | NSPDK↓ | Val.↑ | Uni.↑ | Nov.↑ | FCD↓ | NSPDK↓ |
| Train vs Ref | 100.0 | 99.99 | N/A | 0.074 | 0.0002 | 100.0 | 99.98 | N/A | 0.201 | 0.0001 |
| GDSS (Jo et al., 2022) | 95.72 | 98.46 | **86.27** | 2.900 | 0.003 | 97.01 | 99.64 | 100.00 | 14.656 | 0.019 |
| MiCaM (Geng et al., 2023) | 99.93 | 93.89 | 83.25 | 1.045 | 0.001 | 100.00 | 88.48 | 99.98 | 31.495 | 0.166 |
| DiGress (Vignac et al., 2023) | 99.00 | 96.66 | 33.40 | 0.360 | 0.0005 | 91.02 | 81.23 | 100.00 | 23.060 | 0.082 |
| CatFlow (Eijkelboom et al., 2024) | 99.81 | **99.95** | - | 0.441 | - | 99.95 | **99.99** | - | 13.211 | - |
| DeFoG (Qin et al., 2025) | 99.26 | 96.61 | 72.57 | 0.268 | 0.0005 | 94.97 | 99.98 | 100.00 | 2.030 | 0.002 |
| HOG-Diff (Huang & Birdal, 2026) | 98.74 | 97.10 | 75.12 | 0.172 | 0.0003 | 98.56 | 99.96 | 99.53 | 1.633 | 0.0015 |
| **MIG-VAE (Ours)** | **100.0** | 95.77 | 67.69 | **0.085** | **0.0002** | **100.0** | **100.0** | 99.97 | 1.255 | 0.0011 |
| **RSG-VAE (Ours)** | **100.0** | 96.82 | 74.72 | 0.110 | 0.0004 | **100.0** | **100.0** | 99.98 | **0.706** | 0.0008 |

| Method | MOSES (Polykovskiy et al., 2020) | | | | | GuacaMol (Brown et al., 2019) | | | | |
|---|---|---|---|---|---|---|---|---|---|---|
| | Val.↑ | Uni.↑ | Nov.↑ | FCD↓ | Filters↑ | Val.↑ | Uni.↑ | Nov.↑ | FCD↑ | KL div.↑ |
| Train vs Ref | 100.0 | 100.0 | N/A | 0.518 | 100.0 | 100.0 | 100.0 | N/A | 96.12 | 99.71 |
| DiGress (Vignac et al., 2023) | 85.7 | **100.0** | 95.0 | 1.19 | 97.1 | 85.2 | **100.0** | **99.9** | 68.0 | 92.9 |
| DisCo (Xu et al., 2024) | 88.3 | **100.0** | **97.7** | 1.44 | 95.6 | 86.6 | 86.6 | 86.5 | 59.7 | 92.6 |
| Cometh (Siraudin et al., 2025) | 90.5 | 99.9 | 92.6 | 1.27 | **99.1** | 98.9 | 98.9 | 97.6 | 72.7 | 96.7 |
| DeFoG (Qin et al., 2025) | 92.8 | 99.9 | 92.1 | 1.95 | 98.9 | 99.0 | 99.0 | 97.9 | 73.8 | **97.7** |
| HOG-Diff (Huang & Birdal, 2026) | 99.7 | **100.0** | 89.1 | 0.94 | 96.7 | 99.1 | **100.0** | 97.0 | 78.5 | 96.9 |
| **RSG-VAE (Ours)** | **100.0** | **100.0** | 97.2 | **0.75** | 97.1 | **100.0** | **100.0** | 99.6 | 80.0 | 94.0 |

through atoms and bonds but fail to capture intrinsic higher-order molecular topology (Guo et al., 2022; Vignac et al., 2023). Despite recent advances in topology-aware generation (Sun et al., 2024; Huang & Birdal, 2025), direct generation of higher-order structures remains elusive. This is primarily due to the prohibitive search space; while graph adjacency modeling requires a computation quadratic in the number of nodes, higher-order structures exacerbate computational demands. Another core challenge lies in the intrinsic coupling across different topological dimensions. For example, when generating faces through an edge–face incidence matrix, some boundary edges may be absent from the corresponding node–edge matrix, leading to structural inconsistencies.

To overcome these issues, we adopt a Grammar Variational Autoencoder (GVAE) (Kusner et al., 2017) scheme tailored to rule-sequence representations derived from HGR. This architecture guarantees that all generated molecules are syntactically valid under the grammar, while its latent space captures semantically meaningful variations in molecular topology. After training, molecular generation proceeds via direct sampling from $z \sim \mathcal{N}(0, I)$, which are then decoded into valid molecular structures using the grammar decoder.

**Results.** Quantitative results are summarized in Tab. 1. Across all benchmarks, HGR-based generators achieve 100% chemical validity while maintaining high novelty and diversity. Beyond validity, the consistently lower FCD and NSPDK values indicate that our samples better match the reference data distribution in both chemical and graph topology space, suggesting that HGR provides an effective inductive bias for distributional fidelity rather than merely enforcing syntactic correctness.

Remarkably, even with a simple GVAE backbone, our MIG-VAE and RSG-VAE surpass many recent baselines and, on several datasets, approach the practical ceiling. This strong performance highlights the advantage of representing molecules with higher-order production rules: by exposing ring systems and mesoscopic motifs as explicit generative units, HGR reduces the burden on the neural generator to rediscover long-range constraints from local bonds alone. At the same time, the fact that multiple baselines already achieve high validity on widely used datasets (for example, QM9 and ZINC250k) suggests that these benchmarks can be overly simple and may mask meaningful methodological progress. This observation motivates RingDiv, which is designed to increase topological diversity and provide a more stringent testbed for higher-order molecular generation.

### 3.2 Molecular Foundation Model with HGR

By pretraining large-scale models on diverse multimodal data through multiple pretext objectives, Foundation models (FMs) acquire domain-general abstractions that can be efficiently transferred to downstream tasks with minimal supervision. In molecular science, however, current molecular FMs largely depend on representations such as SMILES strings (Ross et al., 2022; Chang & Ye, 2024) or molecular graphs (Méndez-Lucio et al., 2024; Son et al., 2025), or on simple combinations of the two (Zhu et al., 2023). Since both of them encode essentially the same atomic connectivity and fail to capture the hierarchical and higher-order patterns intrinsic to molecules, it is unlikely that FMs based on these classical representations can exhibit genuinely emergent properties.

To address these gaps, we propose a HGR–based molecular foundation model (HGR-FM) that is both efficient (by leveraging a compact sequence-style encoding) and expressive (by explicitly mod-

eling hierarchical higher-order patterns). Specifically, each molecule is serialized as a rule sequence $\boldsymbol{\pi}^{(i)}$, which is encoded by a Grammar Transformer to produce contextualized representations for all rule tokens. The resulting atom-level token embeddings can be used directly for atom-level tasks. Since the rule sequence includes a molecule-level token, we use its final hidden state as a fixed-length molecular fingerprint. To improve extrapolation, each rule token is initialized with a structure-aware feature that combines Weisfeiler–Lehman (WL) label histograms across refinement rounds with a random-walk structural signature (Dwivedi et al., 2022; Zeng et al., 2024), yielding a permutation-invariant initialization that generalizes to unseen structural patterns.

**Multi-level pretext tasks.** To improve pretraining for molecular FMs, we adopt a multi-level pretext framework (Son et al., 2025) spanning four complementary levels: atom, fragment, molecular, and 3D conformation. The atom-level task reconstructs masked atomic tokens to capture fine-grained local chemical environments. The fragment-level task predicts binary fingerprints, including 2048D Morgan (Rogers & Hahn, 2010) (with a radius of 2) and 166D MACCS (Durant et al., 2002). The molecular-level task regresses molecular descriptors (e.g., molecular weight, logP), encouraging the model to internalize global physicochemical semantics. Finally, at the 3D level, it predicts pairwise interatomic distances from optimized conformers to incorporate spatial geometry.

**Full fine-tuning and probing.** After pretraining, we evaluate our models across a range of unseen downstream tasks using two standard protocols: full fine-tuning and probing. In full fine-tuning, we unfreeze the pretrained encoder and jointly update both the encoder and task-specific prediction heads. While in the probing mode, the pretrained encoder is frozen and only lightweight task-specific probe heads are trained. This setup isolates the quality of learned representations while preventing overfitting on small downstream tasks.

**Results.** We evaluate the performance using the area under the receiver operating characteristic curve (AUC) against established baselines, including Infomax (Veličković et al., 2019), EdgePred (Hamilton et al., 2017), AttrMasking, ContextPred (Hu et al., 2020), GraphCL (You et al., 2020), GraphLoG (Xu et al., 2021), GraphMAE (Hou et al., 2022), GraphMVP (Liu et al., 2022), and MORE (Son et al., 2025). Across seven benchmark datasets from MoleculeNet (Wu et al., 2018), HGR-FM achieves the highest average AUC in both settings (Fig. 2). Re-

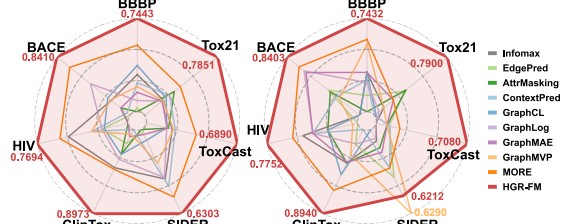

Figure 2: **Performance of HGR-based molecular foundation model.** Radar plots show the normalized AUC of the model pretrained on the ZINC2m under probing (left) and full fine-tuning (right).

markably, the probing performance approached or even surpassed that of full fine-tuning mode. This phenomenon can be partially attributed to catastrophic forgetting introduced by full fine-tuning; however, more importantly, it provides strong evidence that the pre-trained encoder already captures semantically rich and transferable molecular embeddings.

## 4 CONCLUSION

We introduce Higher-order Grammar Representations (HGR), a topology-aware and efficient molecular representation that lifts molecular graphs to combinatorial complexes and serializes each molecule as a compact production-rule sequence. We instantiate three lifting strategies (MEG, MIG, and RSG) to control rule-corpus growth and better accommodate ring systems, and release RingDiv, a ring-enriched dataset of 1.18M molecules with a curated 300k development subset to stress-test higher-order generalization. Extensive experiments demonstrate the utility of HGR for both generation and representation learning. For de novo generation, HGR enables grammar-constrained latent-variable models with 100% theoretical validity and strong distributional fidelity. For representation learning, we develop an HGR-based foundation model that learns from rule sequences alone and injects higher-order structural inductive biases into attention, achieving strong transfer on MoleculeNet under probing and fine-tuning; ablations confirm the central role of higher-order rule tokens. Promising directions include scaling grammar induction and improving rule reuse while controlling corpus growth, integrating 3D conformational signals into the grammar, extending HGR to retrosynthesis and molecular optimization, combining HGR with large language models (LLMs), and further probing interpretability.

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
