# OpenReview forum: "Higher-order grammar representations for molecular generation and learning"
_ICLR.cc/2026/Workshop/LMRL — ICLR 2026 Workshop LMRL Poster_

### Official Review · Reviewer_guyA · 2026-02-24
**The method outlined is promising, however the authors should expand on the presentation of their method, and need to include references and comparisons to other work using hierarchical encoding methods for molecular graphs.**

**Rating:** 3
**Confidence:** 4

**Review:**

This paper presents a novel representation framework for molecules whereby a hierarchical grammar is learnt with which molecules can be encoded and decoded.

In summary the method outlined is promising, however the authors should expand on the presentation of their method, and need to include references and comparisons to other work using hierarchical encoding methods for molecular graphs.

Comments:
* The limitations of SMILES and graphs are well outlined, and it is clear that the proposed hierarchical representation generation framework addresses these limitations.
* Some of the presented results are promising. However, given the comparable results on several metrics (Table 1), the statistical significance of the results should be reported.
* Missing references and a lack of comparison to relevant prior art weaken the authors' claims. In particular, hierarchical representation methods, such as "Hierarchical Generation of Molecular Graphs using Structural Motifs" by Jin et al, 2020 are extremely relevant, having used a similar approach of gradually encoding the molecular graph at different scales, and should have been included in the benchmark.
* Furthermore, the novel "grammar" component and deep learning experiments are insufficiently explained, leading to confusion surrounding the method. In particular, an explicit example accompanying equations 1 and 2 would be illuminating, as well as some details of the training pipeline, model size, and embedding size of the rule tokens the grammar generates. The inclusion of an algorithm describing how the representation is generated would also be beneficial.
* The authors introduce a new benchmarking dataset, which would be an excellent contribution, however the dataset is only very briefly described, and the authors do not report any statistics or performance metrics on this dataset, and have not provided access to this set for review. Similarly, the authors have not reported any ablations, despite mentioning these, making it impossible to disentangle which component of the "higher order grammar" yields the improved performance.
* As can be seen from the authors' presented results, molecule validity has been solved by other work, and they do not demonstrate significant or consistent improvement over other metrics, in particularly the novelty of molecules generated on the QM9 set is low compared to other methods.
* The subgraph matching method to determine the applicability of grammar rules likely scales poorly with molecule size, a known problem for molecule representations, and the authors should therefore compare performance across different molecule sizes. It is also not obvious that the subgraph matching method would handle structural symmetries in molecules well, and the authors should clarify what they mean by "positional encodings tailored for tree structures".
* The results of "probing" and "finetuned" HGR-FM look promising, as a minor comment the radar plots should be normalised between 0 and 1. Better description of the grammar generation algorithm would improve confidence in these results!
* The authors should describe all used acronyms.

---

### Official Review · Reviewer_Rbzb · 2026-02-25
**Sophisticated hierarchical molecular representation technique that allows for de novo generation and molecular representation learning**

**Rating:** 7
**Confidence:** 4

**Review:**

**Summary:**
The paper proposes a new hierarchical representation for molecules that maps the molecular topology into a set of tokens. These representations are used together with a grammar-constrained VAE for de novo generation and for finetuning/probing for molecular representation learning. Moreover, they introduce RingDiv, a ring-enriched benchmark with broader coverage of diverse ring systems.

The proposed novel molecular representation method is compelling and shows promising results. The paper is overall well-written and relatively clear. There are some aspects mentioned later that could further improve the quality of the paper.

In particular, there appear to be two different architectures used for generation and property prediction, respectively. It would be great to discuss these architectures in more detail and clarify how they differ.

**Pros:**
- The presented molecular encoding method is compelling and novel.
- The technique is well explained and visualized.
- The paper reports strong results for molecular generation compared to established baselines. For molecular property prediction, performance also looks promising, albeit for a limited set of benchmarks.

**Areas of improvement:**
- It would be good to add more information about the Grammar-VAE and motivate the choice. For example, one could discuss why it is superior to other token-sequence-based generative models.
- Figure 2 is hard to read because only the outermost line has the explicit performance attached, while there is no indication what score the dashed circles represent. A table with explicit numbers in the appendix could help clarify.
- Presentation of RingDiv: This contribution is not explained sufficiently and the data is not provided with the submission. It appears that it isn’t used anywhere at the moment. Is it used for evaluation or training?


**Further comments:**
- The authors could elaborate more on related work. In the context of molecular foundation models, MolGPS [1] comes to mind (it is also evaluated on some of the probing tasks). For de novo generation, SAFE [2] provides another sequential encoding technique alternative to SMILES that can be used together with an LLM for molecular generation.
- As molecular fingerprints (e.g., MACCS) are used for pretraining in Section 3.2, it would be natural to train MLP probes based on such fingerprints as a baseline.


[1] Sypetkowski, et al. “On the scalability of gnns for molecular graphs” NeurIPS 2024

[2] Noutahi, et al. “Gotta be SAFE" Digital Discovery 2024

---

### Official Review · Reviewer_s7C8 · 2026-02-26
**HIGHER-ORDER GRAMMAR REPRESENTATIONS FOR MOLECULAR TOPOLOGY GENERATION AND UNDERSTANDING**

**Rating:** 8
**Confidence:** 4

**Review:**

The paper is well-structured and technically rigorous, with clear definitions of combinatorial complexes, lifting operations, and grammar induction. The experimental design is comprehensive, comparing HGR-based models with state-of-the-art baselines on multiple benchmarks (QM9, ZINC250k, MOSES) and validating both generative and representation learning capabilities. Visualizations (e.g., Fig 1) effectively illustrate the HGR framework, and the mathematical formulation of production rules and derivation processes is precise. Minor clarity issues exist in the detailed implementation of the three lifting variants (MEG/MIG/RSG) – the paper describes their core ideas but lacks a side-by-side comparison of their rule corpus size and computational overhead on RingDiv.

Cons:
The detailed implementation of MIG/RSG lifting (e.g., anchor-resolving boundary reduction, ring-system guided merging) is underdescribed; a step-by-step example or pseudocode would improve reproducibility.
The generalization of HGR to ultra-large molecular graphs (e.g., macrocycles, polymers) is not evaluated – the paper focuses on small/medium molecules, and it is unclear how HGR performs on topologically extremely complex structures.

---

### Meta-Review · Area_Chair_aoiJ · 2026-02-27

**Recommendation:** Accept (Poster)
**Confidence:** 4

**Metareview:**

Accept.

---

### Decision · Program_Chairs · 2026-03-02

**Decision:**

Accept (Poster)

**Comment:**

Please see the meta-review.